# Mercury Content in Three Edible Wild-Growing Mushroom Species from Different Environmentally Loaded Areas in Slovakia: An Ecological and Human Health Risk Assessment

**DOI:** 10.3390/jof7060434

**Published:** 2021-05-29

**Authors:** Lenka Demková, Július Árvay, Martin Hauptvogl, Jana Michalková, Marek Šnirc, Ľuboš Harangozo, Lenka Bobuľská, Daniel Bajčan, Vladimír Kunca

**Affiliations:** 1Department of Ecology, Faculty of Humanities and Natural Sciences, University of Prešov, 17. Novembra 1, 081 16 Prešov, Slovakia; lenka.bobulska@unipo.sk; 2Department of Chemistry, Faculty of Biotechnology and Food Sciences, Slovak University of Agriculture in Nitra, Tr. A. Hlinku 2, 949 76 Nitra, Slovakia; julius.arvay@gmail.com (J.Á.); marek.snirc@uniag.sk (M.Š.); lubos.harangozo@uniag.sk (Ľ.H.); daniel.bajcan@uniag.sk (D.B.); 3Department of Suitable Development, Faculty of European Studies and Regional Development, Slovak University of Agriculture in Nitra, Tr. A. Hlinku 2, 949 76 Nitra, Slovakia; martin.hauptvogl@uniag.sk; 4Department of Geography and Applied Geoinformatics, University of Prešov, 17. Novembra 1, 081 16 Prešov, Slovakia; jana.michalkova@unipo.sk; 5Department of Applied Ecology, Faculty of Ecology and Environmental Science, Technical University in Zvolen, Ul. T. G. Masaryka 24, 960 01 Zvolen, Slovakia; kunca@tuzvo.sk

**Keywords:** *Imlera badia*, *Boletus subtomentosus*, *Xerocomellus chrysenteron*, mercury, bioaccumulation factor, target hazard quotient

## Abstract

Three edible mushroom species (*Imlera badia*, *Boletus subtomentosus*, *Xerocomellus chrysenteron*) sampled in 60 different localities of Slovakia were evaluated to determine health risks (associated with mercury content) arising from their consumption. Total content of mercury in soil and mushroom samples was determined by an AMA-254 analyzer. Soil pollution by mercury was evaluated by contamination factor (*C_f_*), pollution load index (*PLI*), and geoaccumulation index (*I_geo_*). Bioaccumulation factor (*BAF*), translocation factor (*Qc/s*), provisional tolerably weekly intake (%PTWI), estimated daily intake (*EDI*), and target hazard quotient (*THQ*) were used to describe and compare uptake and transition abilities of mushrooms, and the health risk arising from mushroom consumption. Mercury content in soil significantly influences the content of mercury in mushroom fruiting body parts. Caps and stipes of *Boletus subtomentosus* were found to be the best mercury accumulator. According to the *EDI*, consumption of the evaluated mushroom species represents a serious threat for adults and children. The lowest values of *THQ* were found for *Xerocomellus chysenteron.*

## 1. Introduction

Mushrooms are a part of national gastronomy in Central Europe, which is why mushroom picking is a traditional and very popular activity in the region [1,2]. The consumption of mushrooms has many health benefits, owing to their chemical and nutritional properties [3]. Mushrooms are additionally considered an excellent prebiotic and a source of natural bioactive substances, which can help treat serious diseases [4]. Its content of vitamins, minerals, and trace elements is also remarkable [5]. However, the whole range of anthropogenic activities, including mining activities, industrial activities, and transport, contribute significantly to the deterioration in soil quality. Polluted soil has undesirable substances that could affect growing plants. Mushrooms are considered very good accumulators of pollution coming from soil, and some wild edible mushroom species have even been found to accumulate a high volume of risk elements from soil that is not seriously polluted [6,7]. It is not only the content of risk elements, but also soil pH, organic matter content, and other soil characteristics that influence the intake of risk elements by mushrooms [8,9]. Accumulation and translocation abilities differ between species, genera, or family of mushrooms [10,11]. Differences between edible mushrooms have also found with respect to the ability to accumulate various risk elements [12].

Much attention has been paid to mercury due to translocation from the soil environment to the mushroom fruiting body, as it is a highly toxic element and can have serious consequences for consumers [13,14]. Exposure to mercury causes a variety of neurological, cardiovascular, immunological, and reproductive problems [15].

The composition and pollution of soils in Slovakia are largely influenced by geochemical anomalies, numerous mining areas, long-term continuing wasteful exploitation of natural resources, and a huge number of environmental loads [16]. So far, more than 2000 environmental burdens have been registered in Slovakia, a large part of which is dangerous [17]. Nevertheless, the collection, processing, and consumption of mushrooms is a national tradition in Slovakia, which implies the need to address the dangers arising from mushroom consumption, especially in the most loaded regions. Research of this extent has not yet been carried out in Slovakia.

The aim of the study was (i) to evaluate mercury pollution in soils in selected localities in Slovakia by contamination factor (*C_f_*) and pollution load index (*PLI*); (ii) to determine and compare the content of mercury in three edible mushroom species, with regard to the differences between caps and stipes, (iii) to evaluate and compare the abilities of different edible mushroom species uptake and, translocate mercury from the soils; (iv) to estimate the health risk arising from mushroom consumption.

## 2. Materials and Methods

The samples of three edible mushroom species *Imlera badia* (*n* = 220), *Boletus subtomentosus* (*n* = 88), and *Xerocomellus chrysenteron* (*n* = 193) (identified according to the taxonomic keys [18]) were sampled during 2014–2019 in the selected localities (*n* = 60) in Slovakia (Figure 1). The sampling sites were selected randomly, according to the actual weather conditions suitable for mushroom growing. The number of samples and species that have been found at individual sampling localities are listed in Table 1. In total, 501 edible mushroom samples with the associated soil/substrate (100 g, 0.00–0.10 m) were collected. Soil/substrate samples were taken from three random places, up to 0.5 m from the mushroom sampling point, and subsequently stored in polyethylene bags. Mushroom samples were cleaned from dirt, leaves, and other debits and stored in ventilated polyethylene boxes. On the same day, under laboratory conditions, the mushrooms were cleaned with deionized water, cups were separated from the stipes, and both parts were cut into slices. Subsequently, the mushroom parts were dried in a laboratory oven with the forced air circulation Memmert UF 110 m (Memmert GmbH & Co. KG, Schwabach, Germany) at 40 °C for ~22 h. Dried mushroom samples were homogenized in the rotary homogenizer IKA A 10 basic (IKA-Werke GmbH & Co. KG, Staufen, Germany) and stored in re-sealable PE bags prior to the analysis. Soil samples were, after sampling, cleaned from roots and debits and subsequently dried under laboratory conditions for 3 weeks. Dry soil samples were homogenized, passed through a 2 mm sieve, and stored in paper bags until analysis. Cold-vapour atomic absorption spectrometry (CV-AAS) using AMA-254 (AlTec spol. s r.o., Prague, Czech Republic) with autosampler ASS- 254 (AlTec spol. s r. o., Prague, Czech Republic) was used to measure the Hg content in mushrooms and the associated soil/substrate samples. Quantitative Hg determination was performed at λ = 253.7 nm. Mercury detection limit was 0.0011 mg·kg^−1^ DW and the limit of quantification was 0.0031 mg·kg^−1^ DW [19]. Two CRM materials (ERM-CC 141: Loam soil (IRMM Geel, Belgium) and ERM-CE 278 k: Mussel tissue (IRMM Geel, Belgium)) were measured to check the quality of the measurement. The measurement of the CRM was carried out six times.

Contamination factor (*C_f_*) was used to determine the level of soil pollution by mercury at individual sampling sites [20]. The *C_f_* was determined as follows: (1)$$C_{f}^{i}=\frac{C_{0-1}^{i}}{C_{n}^{i}}$$
where, $C_{0-1}^{i}$ is the content of Hg in the soil samples and $C_{n}^{i}$ is the background value of the Hg in the soils. According to Šefčík et al. [21], the background level of mercury was set to 0.08 mg·kg^−1^ DW. The *C_f_* values were according to Hakanson [20] divided into 4 categories: low contamination factor (*C_f_* < 1), (ii) moderate contamination factor (1 ≤ *C_f_* < 3); (iii) considerable contamination factor (if 3 ≤ *C_f_* < 6), and (iv) very high contamination factor (6 ≥ *C_f_*).

To comprehensively express the level of soil pollution by mercury at each sampling locality, *PLI* was used [22]. *PLI* was calculated as follows: *PLI* = (*C_f_*_1_ × *C_f_*_2_ × *C_f_*_3_ × … × *C_f_*_n_)1/n(2)
where, n is a number of sites in the sampling locality, *C_f_* is a mercury contamination factor belonging to the sampling site within the sampling locality. According to the Wang (2010) [23] four classes of *PLI* were determined: (i) no pollution (*PLI* < 1), (ii) moderate pollution (1 ≤ *PLI* < 2), (iii) heavy pollution (2 ≤ *PLI* < 3), (iv) extreme pollution (*PLI* ≥ 3). 

To quantify the level of contamination on the sampling localities, the index of geoaccumulation was used. The index was calculated as follows:*I_geo_* = log_2_ (*Cn*/1.5*Bn*)(3)
where, *Cn* represents the concentration of mercury in the soil/substrate sample and *Bn* is the geochemical background value (median) for mercury [21]. The background value for mercury was set to 0.08 mg·kg^−1^ DW. According to Müller [24], seven categories of geoaccumulation index were determined: background values *I_ge_* ≤ 0; uncontaminated 0 < *I_geo_* < 1; uncontaminated or slightly contaminated 1 ≤ *I_geo_* ≤ 2; slightly contaminated 2 ≤ *I_geo_* < 3; moderately contaminated 3 ≤ *I_geo_* < 4; strongly contaminated 4 ≤ *I_geo_* < 5; very strongly contaminated *I_geo_* ≥ 5.

To determine the ability of mushrooms to uptake the mercury from soil/substrate to their above-ground parts, bioaccumulation factor (*BAF*) was determined. The bioaccumulation factor of the studied mushrooms species was calculated as follows: (4)$$BAF=\frac{Hg_{m}}{Hg_{s}}$$
where, *Hg_m_*, is the content of Hg in mushrooms, and *Hg_s_* is the content of Hg in soil/substrate (mg·kg^−1^ DW). The *BAF* was calculated separately for mushroom fruit body parts (caps and stipes). The results of *BAF* were divided into three categories: *BAF* < 1 indicates excluder species, *BAF* > 1 indicates accumulators and *BAF* = 1 indicates indicator species [25,26,27].

The translocation quotient (*Qc/s*) expresses the ratio of Hg concentration in cap and stipe and is calculated as follows: (5)$$Q_{c/s}=\frac{Hg_{cap}}{Hg_{stipe}}$$
where, *Hg_cap_* is the concentration of mercury in mushroom caps, and *Hg_stipe_* is the mercury concentration in mushroom stipes. 

Provisional Tolerable Weekly Intake (PTWI) was used to consider the potential risk arising from long-term edible mushroom consumption. According to the WHO [28], PTWI was established to 0.004 mg·kg^−1^ body weight (*BW*) for mercury (0.28 mg/adult person). The %PTWI was determined as follows: (6)$$\%\mathrm{PTWI}=\frac{\left(\mathrm{Hg}\;\mathrm{in}\;\mathrm{mushroom}\;\times \;\mathrm{Intake}\right)}{\mathrm{PTWI}\;\left(\mathrm{Hg}\right)}\times 100\;\left(\%\right)$$
where, Hg in mushroom is the concentration of Hg (mg·kg^−1^ fresh weight (FW)) in the mushroom fruiting body part, Intake stands for the consumption of the studied mushrooms (kg/week FW), PTWI (Hg) = 0.28 mg/adult person. The values exceeding 100% can be considered potentially hazardous. Fresh weight of the mushrooms was calculated providing that the dry matter represented 10% of the mushroom fruiting body [29]. According to the Statistical Office of the Slovak Republic [30], the average amount of consumed “other vegetables and mushrooms” was set to 0.23 kg fresh weight per week.

To point out the transition of Hg to the human body through mushroom consumption, the commonly used estimated daily intake (*EDI*) index was used in this study. *EDI* was calculated as follows: (7)$$EDI=\frac{ADC/C_{E}}{BW}\;\left(\mu g/\mathrm{day}/\mathrm{kg}\right)$$
where, *ADC* is the average daily consumption of mushrooms, which was according to the Statistical Office of the Slovak Republic [30] estimated to be 33 g/day, and *C_E_* is the average Hg concentration in mushroom samples. With the purpose of a comprehensive assessment of the dangers arising from the long-term consumption of mushrooms, the target hazard quotient (*THQ*) was used. *THQ* considered numerous parameters, which can influence the health of consumers [31]. *THQ* can be expressed as the ratio of toxic element exposure and the highest reference dose at which no adverse effects to human health are expected. *THQ* was calculated as follows:(8)$$THQ=\frac{\left(Efr\times ED\times ADC\times C_{E}\right)}{RfDo\times BW\times ATn\;}\times 10^{-3}$$
where, *Efr* is the frequency of exposure (365 days), *ED* is exposure duration (70 years), *ADC* is an average daily consumption of fresh mushrooms (33 g/day), *C_E_* is average Hg concentration in mushroom samples (mg·kg^−1^ FW), and *RfDo* is the oral reference dose for mercury (0.0003 mg/kg/day) [32]. BW is the average body weight (70 kg) and *ATn* is average exposure time (365 days ∗ 70 years = 25,550 days), 10^−3^ is a factor considering the unit’s conversion. If the *THQ* is lower than 1, non-carcinogenic health effects are not expected; if the *THQ* is bigger than 1, there is a serious possibility that adverse health effects can be experienced. 

Open-source Geographic Information System using QGIS (version 2.18) software was used for map outputs processing. Open data from Open Street Map contributors were used [33]. All statistical analyses were performed in R studio [34]. The Mann-Whitney U test was used to determine differences in *BAF* values and PTWI between caps and stipes of individual mushroom species. The Kruskal-Wallis test was used to determine differences in Hg content and *BAF* values between mushroom species. Spearman correlation coefficient was used to determine the relationship between mercury content in the soil, and mushroom fruiting body parts. 

## 3. Results and Discussion

### 3.1. Soil Pollution by Mercury

In total, 501 samples of three edible mushroom species were collected at 60 localities in Slovakia. Associated soil/substrate was collected with the mushroom samples, because it has been confirmed that soil/substrate is the major source of risk elements absorbed via mycelia and rhizomorphs to the mushrooms fruiting body [1]. The content of mercury in soil/substrate ranged between 0.01 to 276 mg·kg^−1^, with an average and median value of 1.95 mg·kg^−1^ and 0.11 mg kg^−1^, respectively. The limit value of mercury for soils in Slovakia [35], which is set to 0.50 mg·kg^−1^ DW, was exceeded at 22 sampling sites, predominantly at former mining areas. These are an important source of risk elements and many of them are in Slovakia classified as environmentally polluted [17]. Contamination of soil samples by Hg was expressed by *Cf*. In Figure 2, the sampling sites are divided into categories according to the level of *Cf* (average *Cf* of sampling sites, for each sampling locality). According to the Hakanson classification [20], from 60 sampling localities, 15 were found as low contaminated, 29 as moderately contaminated, 11 as considerably contaminated, and 5 as very high contaminated by mercury. 

Mercury soil contamination in Slovakia has received attention in earlier studies. Serious pollution has been confirmed in the vicinity of former mining areas, to which utmost attention has been paid in connection with mercury pollution [36,37,38]. The concentrations of mercury that exceeded the permissible limit values were detected in forest soils in Slovakia, also at localities that were not influenced by anthropogenic pollution sources, and geogenic origin of mercury was also excluded [39]. Numerous studies have confirmed the long-distance transmission of air pollution and its serious consequences on environmental quality, even at localities without direct pollution sources [40]. The PLI (Figure 3) was used to give comprehensive information about pollution in the sampling locality. Based on the non-parametrical Kruskal-Wallis test, the *Cf* was significantly higher (*p* < 0.05) in soils where the *I. badia* were sampled, comparing *X. chrysenteron* sampling sites. 

The level of mercury pollution in soil/substrate was also evaluated by the index of geoaccumulation (Igeo). The obtained results showed that at more than half of the sampling sites (*n* = 33), the level of mercury reached background values. Seventeen sampling localities can be considered uncontaminated, 7 uncontaminated or slightly contaminated, only one sampling locality was found as moderately contaminated and two strongly contaminated. Šefčík et al. [21] who used the geoaccumulation index to study the contamination of Slovak soils found that the most serious pollution was linked to mining activities and related industrial activities.

### 3.2. The Content of Mercury in Mushrooms

Protecting human health in terms of food security has gained serious attention worldwide [41,42]. The presence of risk elements such as arsenic, zinc, or mercury in higher amounts in edible mushrooms presents a serious threat for consumers. Consumption of mushrooms has a long tradition in Slovakia, which follows from a historical aspect due to the frequent use of mushrooms in national gastronomy [43,44], which is why the assessment of the risks associated with their consumption is appropriate [15]. The content of Hg in the mushroom body is influenced by substrate characteristics. It has been found in earlier studies that some mushroom genera such as Calocybe, Agaricus, Lepista, Macrolepiota, Boletus, and Lycoperdon tend to accumulate higher amounts of mercury, although growing in less-polluted areas [13]. The content of mercury determined in three edible mushroom species are listed in Table 2. The highest values of mercury were determined in the fruiting body parts of *I. badia*, where the highest values reached 23.7 mg kg^−1^ in the caps and 14.8 mg kg^−1^ in the stipes. Compared with the earlier studies performed in Slovakia, Árvay et al. [45] determined the content of Hg in *I. badia* mushroom species in the range of 5–7 mg kg^−1^. The concentration of mercury in the *X. chrysenteron* collected in Italy ranged between 0.08–0.63 mg kg^−1^ in caps and 0.03 to 0.360 mg kg^−1^ in stipes [26]. This study also focused on the relationship between the substrate and the mushroom body parts in terms of Hg concentration. The results of Spearman’s correlation coefficient are listed in Table 3. The results of the correlation that was made separately for each mushroom species and mushroom body parts showed that there is a strong correlation between all three evaluated parts for all three evaluated mushroom species. The strongest relationship was in all cases confirmed between mushroom parts (cap–stipe), indicating that if the mushroom accumulates risk elements from the environment, all mushroom body parts are affected. The results confirmed a significant relationship between soil pollution by Hg and the content of Hg in mushroom body parts, which is in accordance with the results of Melgar et al. [10], who confirmed that mushroom species accumulated mercury in relation to the underlying soils. Additionally, some authors found that metal concentration in younger mushroom fruiting bodies is higher than in adult ones [46], which can explain the differences between species. The results of the Kruskal-Wallis test, which express the differences in Hg content in mushroom body parts between species, are listed in Table 2. The results showed that in the samples of *I. badia*, Hg content in caps (*p* < 0.01) and in stipes (*p* < 0.05) was significantly higher comparing *X. chrysenteron*.

### 3.3. Bioaccumulation Factor (*BAF*) and the Translocation Quotient (Qc/s)

The bioaccumulation factor was used to express the content of Hg in mushrooms regarding the pollution of the environment (soil/substrate). As mentioned above, soil quality significantly affects the content of risk elements in the different fruiting body parts; the difference lies only in the level of accumulation. The results of *BAF* (Hg) of caps and stipes of three edible mushroom species are listed in Figure 4. Mushroom (and/or plant) is considered an accumulator or hyperaccumulator biosystem if *BAF* > 1 [47]. In the case of *I. badia*, 53% of *BAF* values determined for caps and 38% of *BAF* values determined for stipes were higher than 1. In the case of B. subtomentosus, 78% of *BAF* values determined for caps and 68% of stipes were higher than 1. Finally, *X. chrysenteron* could be considered as an accumulator in 73% of cases determined in caps and 55% of cases determined in stipes. Higher *BAF* values in caps comparing stipes were determined in several earlier studies [1,6]. In the case of *I. badia,* three samples reached *BAF* values lower than 0, which are shown in Figure 4 as outlying values. Those can be considered as bioexcluders. The results of the non-parametrical Mann-Whitney U test showed that for all evaluated species, there was confirmed significantly (*p* < 0.01) higher *BAF* values in caps comparing stipes (Table 4). The distribution of individual mushrooms in Slovakia is different. Among the species evaluated, *I. badia* has the greatest distribution [16] and was therefore found in various types of habitats (from less polluted to relatively clean). This explains the largest dispersion of bioaccumulation factor values. Based on the non-parametrical Kruskal-Wallis test, no significant differences were found between species in *BAF* values determined in stipes and in *BAF* values determined in caps. The translocation quotient (*Qc/s*) in the fruiting body of mushrooms is expressed by the ratio cap/stipe [48] and represents the mobility of each metal in the fruiting body of mushrooms. *Qc/s* values determined for *I. badia* ranged between 0.10–30.1, with an average value 1.99. *Qc/s* values for B. subtomentosus ranged between 0.25–3.82, with an average value 1.54, and *X. chrysenteron Qc/s* values ranged between 0.22–14.3 with an average value 1.73 (Figure 4). Translocation quotient values higher than 1 mean that the concentration of Hg in the caps of evaluated mushrooms is higher than the concentration of Hg in the stipes. High mobility of heavy metals in the fruiting body has the risk of bioaccumulation of high quantities from these elements in the superior part of mushrooms, the part that is more often consumed by humans [49]. In this study, for 99.9% of evaluated mushroom samples (confirmed for all three mushroom species) *Qc/s* values were higher than 1.

### 3.4. Provisional Tolerable Weekly Intake (PTWI)

The health of consumers can be adversely influenced when toxic metals are present in the consumed food [50]. The safe levels of metals intake are determined in terms of provisional tolerable daily intake (PTWI), which is set by the Food and Agriculture Organization/World Health Organization [28]. The results obtained by the analysis of three mushroom species indicate that the average value of %PTWI (regardless of the species) estimated on the Hg content ranged between 0.07–195 %PTWI in the case of an adult consumer, and between 0.15–390 %PTWI in the case of a child consumer (Table 5). The level of 100% PTWI was exceeded only in the case of *I. badia*, but the number of samples is negligible (specifically less than 4% of the caps and less than 0.5% of the stipes, applied for adults and children). The consumption of *B. subtomentosus* and *X. chrysenteron* can be, regarding %PTWI, considered safe. Similar results were determined in an earlier study in Slovakia, where the PTWI of mercury was exceeded only in one mushroom species (*M. procera*), while 15 other mushroom species were evaluated as safe [5]. Comparing mushroom parts, for all three evaluated mushroom species, Hg content in the caps contributed significantly more to the PTWI (*p* < 0.001 in the case of *I. badia* and *X. chrysenteron*, and *p* < 0.05 in the case of *B. subtomentosus*). It was also found that *I. badia* contributed significantly more to the PTWI, comparing *X. chrysenteron* (*p* < 0.001). This has been confirmed in the case of caps, and for stipes. No significant differences in PTWI contribution were found between *I. badia* and *B. subtomentosus* and between *X. chrysenteron* and *B. subtomentosus* (Table 5).

### 3.5. Estimated Daily Intake (EDI)

There is a lot of research on the effects of mercury on human health [15,51,52]. The estimated daily intake (*EDI*) of mercury for adults and children differs between mushroom species. The results of *EDI* determined for *I. badia* ranges between 0.02–26.9 mg/kg/day for adults and 0.05–53.9 mg/kg/day for children; for *B. subtomentosus* ranges between 0.29–11.8 mg/kg/day for adults and 0.58–23.5 for children, and in the case of *X. chrysenteron* ranges between 0.29–50.7 mg/kg/day for adults and 0.59–101.4 mg/kg/day for children. In some earlier studies, *PTDI* (potential tolerable daily intake) was used as a comparative value to identify the danger arising from the consumption of mushrooms expressed by *EDI* [53]. The *PTDI* was established to 0.0005 mg/kg/day for adults (0.035 mg/adult and 0.175 mg/child) [28]. Based on the results obtained, there is a serious threat for adults and children if the evaluated mushrooms were consumed. 

### 3.6. Target Harazd Quotient (THQ)

The target hazard quotient (*THQ*) is frequently used as a marker of human health risk coming from different food consumption because it connects the element concentrations in food with their toxicity, quantity, and quality of food consumption, and consumers’ body mass [53]. The target hazard quotient for mercury determined for three mushroom species was in the range 0.02–26.0 for *I. badia;* 0.06–2.53 for *B. subtomentosus* and (0.01–2.93) for *X. chrysenteron*. The mushroom samples with a *THQ* lower than 1 represent no significant health risk from oral exposure [54]. The consumption of the mushroom samples with *THQ* higher than 1 could represent a health risk for consumers. In the case of *I. badia, THQ* reached values higher than one in 15% of the samples. *THQ* values higher than one were determined for 13% and 6% of the samples of *B. subtomentosus* and *X. chrysenteron,* respectively. It follows that consumption of *X. chrysenteron* can be considered safest compared to the other evaluated species. If the consumers avoid consuming mushrooms from explicitly environmentally polluted areas (mining areas, industrial areas, etc.), consumption of mushrooms in Slovakia in terms of human health risk can be considered safe. We also focused on the general assessment of the sampling sites depending on target hazard quotient. In Figure 5 the level of *THQ* regardless of the mushroom species, is listed.

## 4. Conclusions

The state of soil mercury pollution is largely reflected in the increased mercury content in caps and stipes of edible-mushrooms. The limit value of mercury was exceeded at 22 sampling locations (out of a total of 60 sites). A strong correlation was found between soil mercury content and mushroom fruiting body parts, while significantly higher values of mercury were found in caps comparing stipes. According to the bioaccumulation factor, *B. subtomentosus* was found to be the best accumulator, while hyperaccumulation was confirmed in 78% of the evaluated caps and 68% of the evaluated stipes. The mobility of mercury in the mushroom fruiting body was expressed by translocation factor, which showed that for 99.9% of all evaluated mushroom species, it was higher than 1 (higher values of mercury in caps comparing stipes). The safe level of mercury intake was expressed by PTWI. 100%PTWI was exceeded only in the case of *I. badia*, but only in nine samples from 220. Based on the target hazard quotient, 13%, 12%, and 6% of the samples of *I. badia, B. subtomentosus,* and *X. chrysenteron*, respectively, could represent a health risk for consumers. Based on the obtained results, we can state that the content of risk elements in soil significantly influences the content of risk elements in the mushroom fruiting body. The results showed that the greatest risk is posed by the consumption of mushrooms found in areas classified as environmentally polluted (for example, former mining areas, industrial areas, busy roads, etc.). Consumption of mushrooms from other localities is considered safe; however, we recommend not to exceed the recommended amounts.

## Figures and Tables

**Figure 1 jof-07-00434-f001:**
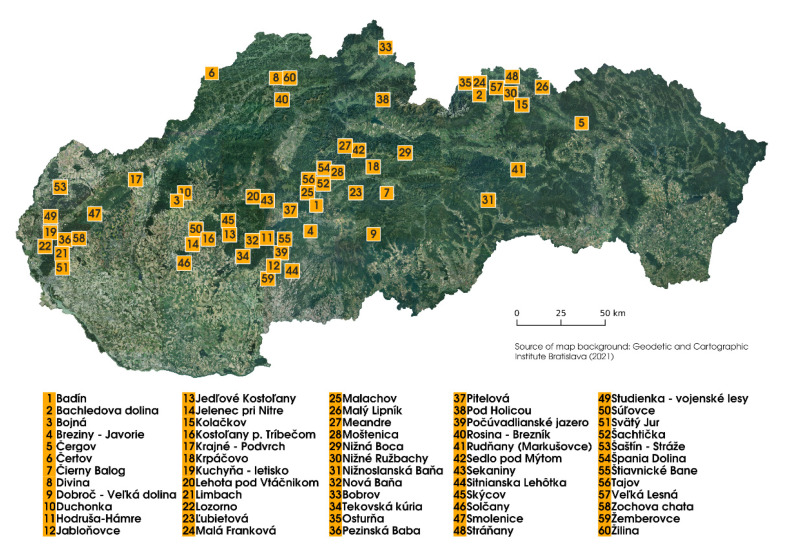
Wild-growing edible mushroom sampling sites.

**Figure 2 jof-07-00434-f002:**
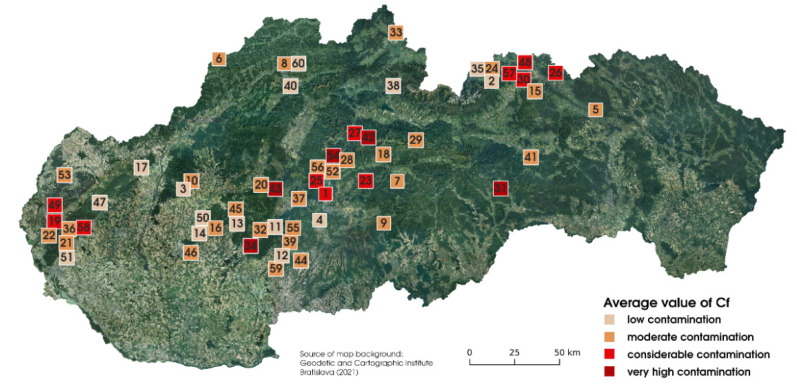
Soil pollution at the sampling localities expressed by contamination factor (*Cf*).

**Figure 3 jof-07-00434-f003:**
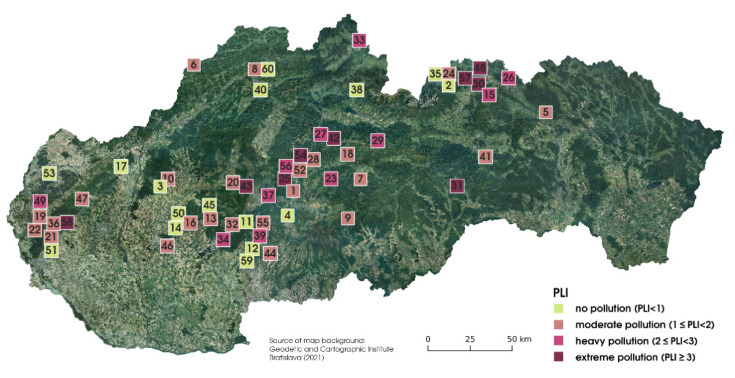
Pollution load index (PLI) values determined for individual sampling localities.

**Figure 4 jof-07-00434-f004:**
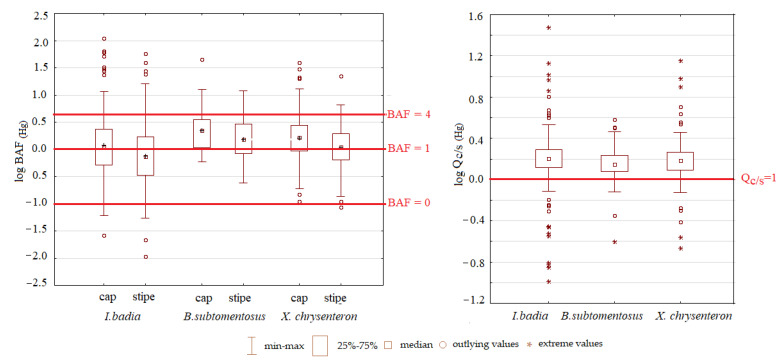
Bioaccumulation factor (BAF) values in caps and stipes and translocation quotient (*Qc/s*) values determined for three edible mushroom species (the data were logarithmized for better visual expression).

**Figure 5 jof-07-00434-f005:**
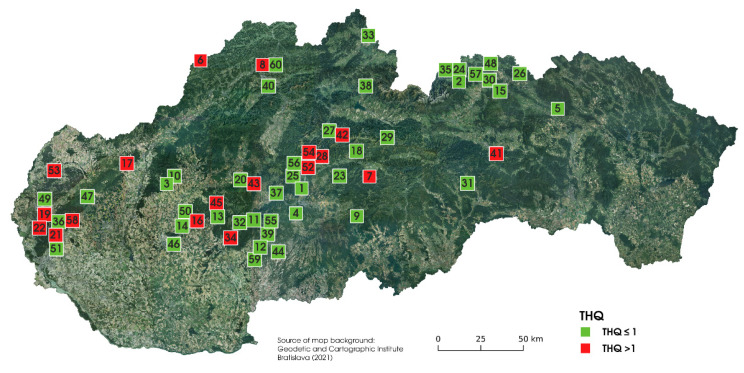
The values of *THQ* at the sampling sites, regardless of the mushroom species.

**Table 1 jof-07-00434-t001:** The occurrence of individual mushroom species in the sampling localities.

No.	Locality	Mushroom Presence and the Number of Samples		
		*I. badia*	*B. subtomentosus*	*X. chystenteron*
1	Badín			4
2	Bachledová Dolina	3		
3	Bojná		5	3
4	Breziny–Javorie		4	
5	Čergov		3	4
6	Čertov	15		
7	Čierny Balog	3		6
8	Divina	3		
9	Dobroč-Veľká dolina	6		
10	Duchonka	6		
11	Hodruša-Hámre			4
12	Jabloňovce	4	4	5
13	Jedľové Kostoľany			3
14	Jelenec pri Nitre		3	4
15	Kolačkov		5	
16	Kostoľany p. Tríbečom	3	5	12
17	Krajné-Podvrch	4	3	13
18	Krpáčovo			12
19	Kuchyňa-letisko	10		
20	Lehota pod Vtáčnikom	8	3	4
21	Limbach			8
22	Lozorno	5		
23	Ľubietová	3	4	4
24	Malá Franková	7		
25	Malachov	4		3
26	Malý Lipník	3	3	
27	Meandre	5		
28	Moštenica	3		4
29	Nižná Boca			6
30	Nižné Ružbachy		4	
31	Nižnoslanská Baňa			3
32	Nová Baňa	8	4	
33	Bobrov	6		
34	Tekovská kúria	4	3	3
35	Osturňa	9		
36	Pezinská Baba		3	5
37	Pitelová	5		
38	Pod Holicou		4	
39	Počúvadlianské jazero			4
40	Rosina-Brezník	4		3
41	Rudňany (Markušovce)	8		
42	Sedlo pod Mýtom	7		
43	Sekaniny	3		3
44	Sitnianska Lehôtka	3		
45	Skýcov		6	16
46	Solčany			3
47	Smolenice	3		14
48	Stráňany	3		
49	Studienka-vojenské lesy	15		
50	Súľovce		5	3
51	Svätý Jur		8	
52	Šachtička	3		6
53	Šaštín-Stráže	11		5
54	Špania Dolina	5		7
55	Štiavnické Bane		3	
56	Tajov	3		3
57	Veľká Lesná	4		10
58	Zochova Chata	12	3	6
59	Žemberovce	6	3	
60	Žilina	3		

**Table 2 jof-07-00434-t002:** Content of Hg (mg kg^−1^) in caps and stipes of three edible mushroom species expressed by descriptive statistics and the results of the Kruskal-Wallis test expressing significant differences in Hg content between species (^a^, *p* < 0.001; ^b^, *p* < 0.05).

Mushroom Species	Cap	Stipe
	Min–Max (Median ± st.Deviation)	
*I. badia*	0.01–23.7 (1.09 ± 3.46) ^a^	0.01–14.8 (0.49 ± 1.45) ^b^
*B. subtomentosus*	0.05–2.59 (0.36 ± 0.47)	0.03–0.99 (0.23–0.23)
*X. chrysenteron*	0.01–2.61 (0.31 ± 0.42) ^a^	0.001–1.04 (0.17 ± 0.17) ^b^

**Table 3 jof-07-00434-t003:** Spearman’s correlation relationships between the substrate and the body parts (cap, stipe) Hg content of three different mushrooms species.

Mushroom Species		Cap	Stipe
*Imlera badia*	substrate	0.54 **	0.51 **
	cap		0.78 **
*Boletus subtomentosus*	substrate	0.57 **	0.55 **
	cap		0.81 **
*Xerocomellus chrysenteron*	substrate	0.32 *	0.48 *
	cap		0.69 **

* *p* < 0.05; ** *p* < 0.01.

**Table 4 jof-07-00434-t004:** The results of the Mann-Whitney U test expressing differences in BAF values between caps and stipes within the species.

Mushroom Species		U	Z	*p*
*I. badia*	between mushroom parts (cap/stipe)	14,428	3.23	0.001 **
*B. subtomentosus*		4118	2.34	0.01 *
*X. chrysenteron*		11,263	4.12	0.001 **

* *p* < 0.05, ** *p* < 0.01.

**Table 5 jof-07-00434-t005:** Percentage of the PTWI estimated on Hg content (min-max (average ± st.deviation)), separately in caps and stipes of three analysed mushroom species, considering the age of the consumer. The results of the Kruskal-Wallis test expressing the significant differences (^d,e^ *p* < 0.001) between species in PTWI contribution and the results of the Mann-Whitney U test expressing significant differences (^a,c^ *p* < 0.001; ^b^ *p* < 0.05) in PTWI contribution between mushroom parts of individual mushroom species.

Mushroom Species	Adult		Child	
	Cap	Stipe	Cap	Stipe
*I. badia*	0.07–195 ^a,d^ (9.01 ± 28.6)	0.11–121 ^a,e^ (4.07 ± 11.9)	0.15–390 ^a,d^ (18.0 ± 57.1)	0.22–242 ^a,e^ (8.15 ± 23.9)
*B. subtomentosus*	0.39–21.0 ^b^ (2.95 ± 3.83)	0.23–8.09 ^b^ (1.89 ± 1.89)	0.79–42.1 ^b^ (5.80 ± 7.66)	0.46–16.2 ^b^ (3.77 ± 3.78)
*X. chrysenteron*	0.08–21.5 ^c,d^ (2.53 ± 3.49	0.07–8.55 ^c,e^ (1.42 ± 1.37)	0.16–42.9 ^c,d^ (5.07 ± 6.98)	0.15–17.1 ^c,e^ (2.85 ± 2.72)

## Data Availability

The data that support the findings of this study are available from the corresponding author upon reasonable request.

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
