# Peer review of "Mercury Content in Three Edible Wild-Growing Mushroom Species from Different Environmentally Loaded Areas in Slovakia: An Ecological and Human Health Risk Assessment"

_jof, 2021, doi:10.3390/jof7060434_

Round 1
Reviewer 1 Report
The current manuscript presents a study based on the determination of mercury content and the consequent health risk assessment of edible mushrooms cultivated in different environmentally loaded areas. Before publishing, the authors should address some topics. More specifically:
1) Line 40: Change ''consisting of'' to ''based on'' or ''which are attributed to''
2) Please enrich the introduction part with some more info regarding the objective of your study. For instance, the authors can add a paragraph with info about the three examined species, or some pollution stats concerning heavy metals in Europe or even Slovakia, or some more reference regarding the studies and the results of mercury determination in mushrooms (what techniques were used, which were the conclusions, etc).
3) Please provide the aim of the study alone in a single paragraph in order to be clear the aim of your work.
4) Explain and discuss more the conclusions of Spearman correlation (Table 3) since a relatively strong relationship of species, body parts and Hg content is only proved in two cases (0.78 and 0.81). The other values in Table 2 does not support a strong relationship relationship.
5) Please make a more detailed discussion for the Box-plots of Figure 4. In which factors do the authors attribute the high dispersion of values in the case of BAF and Qc/s in I. badia or in the Qc/s of X. chrysenteron?
6) Line 299: Add ''are present'', not ''present''
7) In Table 5, add the letters for significant differences also in the results forchildren's mushroom consumption.
8) Section 3.4 and section 3.5 seems to provide contradictory results. According to Section 3.4 ''The consumption of B. subtomentosus and X. chrysenteron can be, regarding the %PTWI, considered safe''. However, based on Section 3.5 '' there is a serious threat for adults and children if the evaluated mushrooms were consumed''. Please comment more on these results.
9) Please make add some more info in the conclusions section. For instance, which of the species is more safe for consumption and why? Which parameters should evaluate or take into consideration a producer for a safe mushroom cultivation and market?
10) Since there are various published works in the literature regarding mercury content in edible mushrooms in different countries, please focus and highlight more the novelty of the present work in the Introduction section.
Author Response
Dear reviewer,
the authors would like to thank you for all recommendations and suggestions, which undoubtedly contributed to improving the quality of this paper. All specific comments are elaborated below. All changes in the manuscript are marked by yellow color.
Reviewer #1
The current manuscript presents a study based on the determination of mercury content and the consequent health risk assessment of edible mushrooms cultivated in different environmentally loaded areas. Before publishing, the authors should address some topics. More specifically:
- Line 40: Change ''consisting of'' to ''based on'' or ''which are attributed to''
Thank you. The sentence was changed according the suggestion.
- Please enrich the introduction part with some more info regarding the objective of your study. For instance, the authors can add a paragraph with info about the three examined species, or some pollution stats concerning heavy metals in Europe or even Slovakia, or some more reference regarding the studies and the results of mercury determination in mushrooms (what techniques were used, which were the conclusions, etc).
Thank you for your comment. The introduction has been expanded.
3) Please provide the aim of the study alone in a single paragraph in order to be clear the aim of your work.
The „aim of the study“ part was separated to the single paragraph.
- Explain and discuss more the conclusions of Spearman correlation (Table 3) since a relatively strong relationship of species, body parts and Hg content is only proved in two cases (0.78 and 0.81). The other values in Table 2 does not support a strong relationship relationship.
Thank you for your comment. Because, almost in all cases the correlation was significant at the level p<0.01, the authors do not consider it necessary to address the differences between species, as these differences are minimal. Another situation would occur if in some cases no significant correlation was found, resp. if the correlation was negative - which did not happen in our case.
- Please make a more detailed discussion for the Box-plots of Figure 4. In which factors do the authors attribute the high dispersion of values in the case of BAF and Qc/s in I. badia or in the Qc/s of X. chrysenteron?
Thank you for your suggestion. The information was added to the manuscript.
6) Line 299: Add ''are present'', not ''present''
Thank you. The sentence was corrected.
7) In Table 5, add the letters for significant differences also in the results forchildren's mushroom consumption.
The letters were added to the Table 5.
8) Section 3.4 and section 3.5 seems to provide contradictory results. According to Section 3.4 ''The consumption of B. subtomentosus and X. chrysenteron can be, regarding the %PTWI, considered safe''. However, based on Section 3.5 '' there is a serious threat for adults and children if the evaluated mushrooms were consumed''. Please comment more on these results.
Thank you for your comment. The PTWI (and also THQ) is more important for the risks arising from the consumption of mushrooms than EDI. Additionally, the PTWI for mercury is defined by WHO. EDI is was used only as additional information because it is primarily computed for macro elements (which are essential for humans). It is clear that in the case of Hg (which is dangerous for the human body) the EDI values will be considered dangerous.
9) Please make add some more info in the conclusions section. For instance, which of the species is more safe for consumption and why? Which parameters should evaluate or take into consideration a producer for a safe mushroom cultivation and market?
Addtional information was added to the conclusion. Thank you for your suggestion.
10) Since there are various published works in the literature regarding mercury content in edible mushrooms in different countries, please focus and highlight more the novelty of the present work in the Introduction section.
Thank you for your suggestion. The novelty was added to the Introduction part.

Reviewer 2 Report
The manuscript is interesting because of the theme it describes, in addition to the importance it has in showing evidence of contamination by heavy metals in edible mushrooms.
The attached file shows the observations made. In general, it is suggested:
- Take care of the format of scientific names. These must be written in italics.
- Attach a section corresponding to the evidence of the traditional consumption of the mushrooms in question, as well as bibliographic sources that support it
- In the methods section indicate how the taxonomic identification of the fungi was carried out. -Include the criteria for choosing the sampling sites.
- According to the results obtained, a section should be included with proposals for solutions to the problem of consuming contaminated mushrooms. Also, an alert should be included for government agencies regarding the problem of contaminated soils and its impact on natural resources.

Author Response
Dear reviewer,
the authors would like to thank for all recommendations and suggestions, which undoubtedly contributed to improving the quality of this paper. All specific comments are elaborated below. All changes in the manuscript are marked by yellow color.
- Take care of the format of scientific names. These must be written in italics.
Scientific names were checked and rewritten. Thank you for your comment.
- Attach a section corresponding to the evidence of the traditional consumption of the mushrooms in question, as well as bibliographic sources that support it
The section was added.
- In the methods section indicate how the taxonomic identification of the fungi was carried out. -Include the criteria for choosing the sampling sites.
Thank you for your suggestion. The reference for taxonomic key was added. The co-author of the study (prof. Kunca) is one of the well-known mycologists in Slovakia. He was responsible for the taxa indentificantion.
- According to the results obtained, a section should be included with proposals for solutions to the problem of consuming contaminated mushrooms. Also, an alert should be included for government agencies regarding the problem of contaminated soils and its impact on natural resources.
The section was added to the Conclusion part.
All additional recommendations and mistakes, which were listed in the manuscript were corrected and incorporated.
Best regards,

Reviewer 3 Report
This researh aimed to estimate Hg contents in three wild edible mushroom species (in total 501 samples), and food safety risk assessment. A variety of caculation models were used in this paper. This will provide a reference for other researchers. However, there are many problems and deficiencies in this paper.
- there were many writing format peoblems, for examples, line30:Boletus subtomentosus should be italics; Line59-60:contamination factor and pollution load index should be abbreviated and the abbreviation should be used in the following texture.
- line51-52: please add the refference Sun LP, Chang WD, Bao CJ, Zhuang YL. Metal contents, bioaccumulation, and health risk assessment in wild edible Boletaceae mushrooms. Journal of Food Science, 2017, 82: 1500-1508.
- please provied the infermations of certified reference materials (CRM) used in Hg
determination and the values determined using certified reference materials.
- according to the calculation formula of %PTWI、EDI、THQ, the concentration of Hg (mg.kg-1 fresh weight (FW)) in the mushroom fruiting body were needed, however, there were no infermations about Hg contents in 501 mushroom samples in paper. Please add the concentrations of Hg (mg.kg-1 FW or mg.kg-1 DW ) in the fruiting bodes of 501 samples.
- according to the calculation formula of EDI, μg/day/kg should be used in the caculation resules, why mg/kg/day were used in lines326-337?please confirm whether the values of line326-339 is correct.
- the format of reference is not uniform, please modify.
Author Response
Dear reviewer,
the authors would like to thank for all recommendations and suggestions, which undoubtedly contributed to improving the quality of this paper. All specific comments are elaborated below. All changes in the manuscript are marked by yellow color.
there were many writing format peoblems, for examples, line30:Boletus subtomentosus should be italics; Line59-60:contamination factor and pollution load index should be abbreviated and the abbreviation should be used in the following texture.
Thank you for your comment. Abbreviations of contamiantion factor and pollution load index were used in the text of manuscript.
line51-52: please add the refference Sun LP, Chang WD, Bao CJ, Zhuang YL. Metal contents, bioaccumulation, and health risk assessment in wild edible Boletaceae mushrooms. Journal of Food Science, 2017, 82: 1500-1508.
Thank you for suggestion. The reference was added to the text.
please provied the infermations of certified reference materials (CRM) used in Hg.
The information about the CRM material was added to the methodology part.
according to the calculation formula of %PTWI、EDI、THQ, the concentration of Hg (mg.kg-1 fresh weight (FW)) in the mushroom fruiting body were needed, however, there were no infermations about Hg contents in 501 mushroom samples in paper. Please add the concentrations of Hg (mg.kg-1 FW or mg.kg-1 DW ) in the fruiting bodes of 501 samples.
Thank you for your comment. As this is a huge amount of data, and for the purposes of the work it was not necessary to present them, the authors decided not to include this data in the work. If you are interested, we can provide them.
according to the calculation formula of EDI, should be used in the caculation resules, why mg/kg/day were used in lines326-337?please confirm whether the values of line326-339 is correct.
Thank you for your suggestion. The results of EDI were recalculated from μg/day/kg to mg/kg/day because it is better interpretable, understendable for readers in this form. Also we would like to compare the results with PTDI values which are also set in mg/kg/day.
the format of reference is not uniform, please modify.
Thank you. References were properly checked and corrected.
Best regards,

Round 2
Reviewer 3 Report
Ok